# ER Stress Induces Cell Cycle Arrest at the G2/M Phase Through eIF2α Phosphorylation and GADD45α

**DOI:** 10.3390/ijms20246309

**Published:** 2019-12-13

**Authors:** Duckgue Lee, Daniel Hokinson, Soyoung Park, Rosalie Elvira, Fedho Kusuma, Ji-Min Lee, Miyong Yun, Seok-Geun Lee, Jaeseok Han

**Affiliations:** 1Soonchunhyang Institute of Medi-bio Science (SIMS), Soonchunhyang University, Cheonan-si, Chungcheongnam-do 31151, Korea; tomyoo27@gmail.com (D.L.); daniel.hokinson@gmail.com(D.H.); ivydawn92@gmail.com (S.P.); maria.rosalie.elvira@gmail.com(R.E.); fedhokusuma@gmail.com (F.K.); jmajw@hanmail.net (J.-M.L.); 2Department of Bioindustry and Bioresource Engineering, College of Life Sciences, Sejong University, Seoul 02447, Korea; myyun91@sejong.ac.kr; 3KHU-KIST Department of Converging Science & Technology, Department of Science in Korean Medicine, and Bionanocomposite Research Center, Kyung Hee Univerisity, 26 Kyungheedae-ro, Seoul 02447, Korea; seokgeun@khu.ac.kr

**Keywords:** GADD45α, ER stress, eIF2α phosphorylation, G2-M cell cycle arrest, cell death

## Abstract

Endoplasmic reticulum (ER) stress is known to influence various cellular functions, including cell cycle progression. Although it is well known how ER stress inhibits cell cycle progression at the G1 phase, the molecular mechanism underlying how ER stress induces G2/M cell cycle arrest remains largely unknown. In this study, we found that ER stress and subsequent induction of the UPR led to cell cycle arrest at the G2/M phase by reducing the amount of cyclin B1. Pharmacological inhibition of the IRE1α or ATF6α signaling did not affect ER stress-induced cell cycle arrest at the G2/M phase. However, when the alpha subunit of eukaryotic translation initiation factor 2 (eIF2α) phosphorylation was genetically abrogated, the cell cycle progressed without arresting at the G2/M phase after ER stress. GEO database analysis showed that growth arrest and DNA-damage-inducible protein α (*Gadd45α*) were induced in an eIF2a phosphorylation-dependent manner, which was confirmed in this study. Knockdown of GADD45α abrogated cell cycle arrest at the G2/M phase upon ER stress. Finally, the cell death caused by ER stress significantly reduced when GADD45α expression was knocked down. In conclusion, GADD45α is a key mediator of ER stress-induced growth arrest via regulation of the G2/M transition and cell death through the eIF2α signaling pathway.

## 1. Introduction

Accumulation of unfolded or misfolded proteins in the endoplasmic reticulum (ER), commonly referred to as ER stress, triggers the unfolded protein response (UPR). The UPR is a protective process that restores cellular homeostasis by controlling protein translation, folding, and degradation in response to ER stress in cells. There are three ER-transmembrane proteins that activate the UPR signaling pathway: PKR-like eukaryotic initiation factor 2 kinase (PERK), inositol-requiring enzyme 1 α (IRE1α), and activating transcription factor 6 α (ATF6α). When PERK is activated by its dimerization, it phosphorylates the alpha subunit of eukaryotic translation initiation factor 2 (eIF2α), which induces rapid and transient translational attenuation [1]. Meanwhile, certain mRNAs, including activating transcription factor 4 (*ATF4*), are paradoxically and preferentially translated [2,3]. ATF4 then enhances transcription of C/EBP homologous protein (*CHOP*), which has a role in the determination of cell fate under stress conditions [4]. On the contrary, the activated IRE1α produces a spliced form of X-box protein 1 (sXBP1), which induces several genes that are involved in chaperoning activities or ER-associated protein degradation (ERAD) through its endonuclease activity [5]. ATF6α is released from the ER to the Golgi apparatus to produce its active form, which acts as a transcription factor, leading to induction of its target genes upon ER stress [6]. These proteins act as a double-edged sword that protects cells from stress either by restoring ER homeostasis or inducing cell death [7].

Cell cycle checkpoints are control mechanisms that ensure the accurate replication of the genome to pass genomic integrity onto the daughter cell [8]. They play an important role in recognizing defects in essential processes including DNA replication or chromosome segregation, and induce cell cycle arrest through regulation of the activities of specific cyclin and cyclin-dependent kinases (CDKs) complexes at different phases in response to DNA damage [9]. Many DNA damage response proteins, including Ataxia telangiectasia mutated (ATM) / Ataxia Telangiectasia and Rad3 related (ATR) or checkpoint kinases 1/2 (Chk1/2), are involved in cell cycle control; therefore, cell cycle checkpoints are typically associated with DNA damage [10,11].

Over the last decade, it has been suggested that ER stress causes cell cycle arrest. G1/S transition arrest upon ER stress is achieved through reducing the amount of cyclin D1 due to repression of its synthesis via eIF2α phosphorylation [12]. In addition to G1/S arrest, it has been reported that ER stress might be involved in cell cycle arrest at the G2/M phase. Several natural compounds and pharmacological reagents cause ER stress-induced G2/M cell cycle arrest and cell death in diverse cell lines [13,14,15]. However, the molecular mechanisms and the role of the UPR underlying G2/M cell cycle arrest still need to be investigated.

The aim of this study was to investigate the effects of ER stress on the G2/M transition and to identify which UPR pathway is involved in this phenomenon using mouse embryonic fibroblasts (MEFs), which are powerful tools to study the UPR [16]. In addition, we analyzed the Gene Expression Omnibus (GEO) database derived from MEF cells exposed to ER stress to identify Growth arrest and DNA-damage-inducible protein α (*Gadd45α*), which is induced through the eIF2α phosphorylation signaling pathway. Furthermore, we carried out a loss-of-function study to discover the role of GADD45α in G2/M cell cycle arrest and cell death upon ER stress.

## 2. Results

### 2.1. ER Stress and Subsequent Induction of the UPR Arrested Cell Cycle Progression at the G2/M Phase

First, we investigated how cell cycle progression would be affected by ER stress. Cell cycle progression of wild type (Eif2α^S/S^) MEFs was observed in the presence or absence of ER stress caused by thapsigargin (TG), a well-known ER stress inducer that works by disturbing calcium homeostasis [16]. Cell cycle progression was arrested at the G2/M phase in the presence of ER stress, but not in control MEFs, at 8 h (Figure 1A). To verify whether the UPR was induced or not, we checked the induction profile of several UPR genes. The expression of spliced *Xbp1* (IRE1α pathway), *Chop* (PERK/eIF2α pathway), *p58* (IRE1α/ATF6α pathway), and *BiP* (IRE1a/ATF6a pathway) were significantly increased at 8 h after TG treatment (Figure 1B). In addition, the levels of ATF4 and CHOP proteins were also highly increased, suggesting that ER stress was induced and the UPR was activated at this time point (Figure 1C). Since cell cycle progression is mediated by a variety of cyclin proteins, we checked the expression levels of various cyclin proteins at 8 h after TG treatment. It is known that cyclin B1 is induced to enter the mitotic axis during the G2/M transition in normal cell cycle progression [17]. In our experiments, the amount of cyclin B1 was significantly diminished in the presence of ER stress at 8 h compared to the control (Figure 1C). Expression of cyclin A, another cyclin protein involved in the G2/M transition through association with Rb and E2F-1 [18], was not significantly changed by the presence of ER stress at 8 h compared to the control (Figure 1C). These results suggest that ER stress induces cell cycle arrest at the G2/M phase by regulating the amount of cyclin B1.

### 2.2. The PERK-eIF2α Pathway Is Involved in G2/M Cell Cycle Arrest

Next, we investigated which signaling pathway of the UPR was involved in cell cycle arrest at the G2/M phase. First, we checked the IRE1α signaling pathway using 4μ8c, which is known to inhibit IRE1α RNase activity [19]. Treatment with 4μ8c significantly inhibited splicing of *Xbp1 (sXbp1)* in a dose-dependent manner, whereas the total amount of *Xbp1 (tXbp1)* was not changed at 8 h (Figure 2A,B). However, there was no significant difference in the pattern of cell cycle progression at the G2/M phase between 4μ8c-treated and control MEFs at 8 h in the presence of ER stress, suggesting that the IRE1α signaling pathway might not be involved in ER stress-mediated cell cycle arrest at the G2/M phase (Figure 2C).

Next, we used ceapinA7, a specific inhibitor of the ATF6α signaling pathway [20]. CeapinA7 efficiently attenuated the induction of *BiP* (Figure 3A), *ERo1l* (Figure 3B), and *Herpud1* (Figure 3B), which are known ATF6α target genes. However, treatment with ceapinA7 did not affect the change in cell cycle progression at 8 h caused by ER stress (Figure 3D), suggesting that the ATF6α signaling pathway is not a key player in ER stress-mediated G2/M cell cycle arrest.

Finally, we checked the PERK-eIF2α signaling pathway using eIF2α Ser51Ala mutant MEFs (Eif2α^A/A^), which are widely used to observe the role of eIF2α phosphorylation under stress conditions [21]. We observed that ER stress did not arrest cell cycle progression at G2/M in Eif2α^A/A^ (Figure 4A). There were increased levels of *BiP* (Figure 4B), spliced *Xbp1* (Figure 4C), and total *Xbp1* (Figure 4D) at 8 h in the presence of ER stress in Eif2α^A/A^, suggesting that the other branches of the UPR were properly induced. Thus, it is likely that the eIF2a phosphorylation signaling pathway is crucial for ER stress-mediated cell cycle arrest at the G2/M phase. Since disturbed calcium homeostasis via TG might affect cell cycle progression [22], we tried to rule out the calcium effect in our experiment. For this, we used MEFs expressing Fv2E-PERK, which can directly phosphorylate eIF2α without ER stress [23,24]. When Fv2E-PERK was dimerized and activated by AP20187, cell cycle arrest at the G2/M phase was significantly observed compared to the control at 8 h (Figure 4E). These results suggested that the PERK-eIF2α signaling pathway, but not the other branches of the UPR, is involved in the cell cycle arrest at the G2/M phase caused by ER stress.

### 2.3. GADD45α Is Induced by ER Stress through the PERK-eIF2α Pathway

Next, we investigated which protein was critical for ER stress-mediated cell cycle arrest. Since we observed that ER stress-induced cell cycle arrest at the G2/M phase through downregulation of cyclin B1 protein, we hypothesized that certain proteins might be induced by ER stress that are responsible for regulation of cyclin B1 protein levels. To find potential ER stress-inducible target genes, we analyzed the public GEO database (GSE35681) [25]. We sorted out significantly upregulated genes in MEFs at 8 h in response to ER stress-induced by treatment with tunicamycin (TM), another well-known ER stress inducer that blocks *n*-linked glycosylation (Appendix A) [16]. Through gene ontology (GO) analysis, we identified nine genes that were involved in regulation of the cell cycle (GO:0051726) (Figure 5A and Appendix A). Among the nine genes, we focused *Gadd45α* since it is known to induce G2/M cell cycle arrest. Notably, it was previously reported that increased GADD45α expression was induced through eIF2α phosphorylation upon arsenite treatment [26]. Based on this, we investigated whether GADD45α is a key player in ER stress-mediated cell cycle arrest at the G2/M phase. First, we checked whether the expression level of GADD45α protein was changed after ER stress. The amount of GADD45α protein increased at 4 h and persisted up to 12 h after TG treatment in Eif2α^S/S^ MEFs, whereas GADD45α expression was not induced in Eif2α^A/A^ MEFs (Figure 5B). The mRNA levels of *Gadd45α* were also significantly increased at 2 h and persisted up to 8 h after TG treatment in Eif2α^S/S^ MEFs, whereas there was no expression change in Eif2α^A/A^ MEFs (Figure 5C). We also used 4μ8c and ceapinA7 to confirm whether eIF2α phosphorylation is crucial for GADD45α induction upon ER stress. Chemical inhibition of IRE1α and ATF6α pathway did not affect the induction pattern of *Gadd45α* upon ER stress (Figure 5D,E). These data suggested that GADD45α was induced in response to ER stress in an eIF2α phosphorylation-dependent manner.

### 2.4. Knockdown of GADD45α Impaired G2/M Cell Cycle Arrest Caused by ER Stress

Next, we investigated whether GADD45α was responsible for the cell cycle arrest at the G2/M phase caused by ER stress. For this purpose, we employed siRNAs to knockdown the expression of GADD45α protein. The induction of GADD45α protein at 8 h after TG treatment was significantly reduced by GADD45α siRNA (Figure 6A). The reduction of cyclin B1 protein levels at 8 h after TG treatment was also restored by GADD45α siRNA, suggesting that GADD45α is a key regulator of cyclin B1 expression upon ER stress (Figure 6A). Next, we investigated whether knockdown of GADD45α protein affected the cell cycle arrest at the G2/M phase in the presence of ER stress. We observed that the cell cycle arrest at the G2/M phase at 8 h after TG treatment was eliminated by GADD45α siRNA (Figure 6B). These data suggest that GADD45α is a key protein for ER stress-mediated cell cycle arrest at the G2/M phase.

### 2.5. Cell Cycle Arrest at the G2/M Phase Caused by GADD45α-Induced Cell Death

Last, we investigated the physiological effect of GADD45α-mediated cell cycle arrest at the G2/M phase upon ER stress. Previous studies have demonstrated that GADD45α causes translocation of BIM to mitochondria and the subsequent release of Cytochrome c to induce apoptosis [27]. Therefore, we hypothesized that GADD45α induction in the presence of ER stress might be linked to cell death. To test whether GADD45α regulates cell death upon ER stress, we checked the expression profile of pro-apoptotic proteins. The expression levels of cleaved forms of Poly (ADP-ribose) polymerase (PARP) and Caspase-3 at 12 h after TG treatment were significantly decreased in GADD45α-knockdown cells (GADD45a KD) compared to scramble-treated cells (Scramble) (Figure 7). We also observed that the expression level of CHOP, a key marker of ER stress-mediated cell death, was reduced by GADD45α knockdown upon ER stress, suggesting that GADD45α is required to induce cell death upon ER stress (Figure 7).

## 3. Discussion

In the current study, we report that cell cycle progression was arrested at the G2/M phase upon ER stress and subsequent induction of the UPR. Among the three UPR pathways, the PERK-eIF2α signaling pathway, but not the IRE1α/XBP1 or ATF6α signaling pathways, was responsible for ER stress-mediated cell cycle arrest at G2/M. We found that GADD45α was induced by eIF2a phosphorylation and acted as a key molecule to control cell cycle progression upon ER stress. Knockdown of GADD45α expression significantly abrogated the ER stress-mediated cycle arrest at the G2/M phase, as well as cell death. These results suggest that ER stress causes cell cycle arrest at the G2/M phase through eIF2α phosphorylation and subsequent induction of GADD45α.

Accumulating evidence has suggested that ER stress causes cell cycle arrest at the G2/M transition [28,29,30]. However, it was not previously clear how ER stress and the UPR signaling pathway were directly involved in controlling the G2/M cell cycle progression. As ER stress promotes activation of the UPR, we investigated which of the three branches of the UPR was involved in the cell cycle arrest at the G2/M phase. Pharmacological approaches using IRE1α or ATF6α inhibitors did not affect ER stress-mediated G2/M cell cycle arrest. In contrast, we observed that genetic mutation of eIF2α diminished ER stress-mediated G2/M cell cycle arrest. In addition, a genetic approach that selectively induces PERK dimerization by Fv2E-PERK without disruption of calcium homeostasis arrested cell cycle progression at the G2/M phase. These results strongly suggested that the PERK-eIF2α signaling pathway might contribute to regulation of the cell cycle checkpoint at the G2/M phase. Because ATF4, a downstream target of eIF2α phosphorylation, was reported to participate in cell cycle arrest at the G1 phase in different cell lines [31], activation of ATF4 should be investigated for its role in cell cycle regulation at the G2/M phase in further studies.

Our analysis of the mouse GEO database revealed that nine genes might be involved in cell cycle regulation in response to ER stress. Apurinic/apyrimidinic endonuclease 1 (APE1) [32] and the E3 ubiquitin ligase TRIM32 [33] have been reported to be regulators of cellular senescence, which is an irreversible cell cycle arrest. Polo-Like Kinases 2 (Plk2) [34] and the transcription factor ATF5 [35] are important for centroid duplication during the G1 phase. Neurofibromatosis Type 2 Protein (NF2) is involved in cell cycle regulation, although it has not been shown to have a direct effect on the cell cycle [36]. Other proteins, including GADD45α, The Aurora Kinase A (AURKA) [37], and Mouse double minute 2 homolog (MDM2) [38], have been shown to regulate cell cycle progression at the G2/M phase. Interestingly, it was reported that induction of GADD45α increased its association with cyclin-dependent kinase 1 (CDK1; also known as cell division cycle protein 2 homolog) and forced it to dissociate from cyclin B1/CDK1 complexes. Free cyclin B1 undergoes proteasome-mediated degradation, leading to G2/M cell cycle arrest [39]. These observations prompted us to further investigate the role of GADD45α in ER stress-mediated cell cycle arrest at the G2/M phase. GADD45α is a member of the p53-dependent or independent DNA damage response (DDR) pathway and has been implicated in diverse processes such as DNA repair, cell cycle progression, and apoptosis through interacting with its partner [40,41,42]. In this study, we observed that knockdown of GADD45α using siRNA impaired the reduction of cyclin B1 proteins upon ER stress. In addition, the cell cycle arrest at G2/M and cell death caused by ER stress was diminished by knockdown of GADD45α in MEFs, which indicated that GADD45α was required for cell cycle arrest at the G2/M phase and cell death upon ER stress.

Finally, GADD45α is frequently dysregulated in cancer, particularly in which lacks the expression of GADD45α [43,44]. Thus, pharmacological agents that target GADD45α enhance the efficiency of radiotherapy [45]. Therefore, anti-cancer approaches consisting of targeting combination of the UPR pathway and GADD45α with anticancer drugs may be helpful for the treatment of cancer with minimizing side effect and chemotherapeutic resistance.

## 4. Materials and Methods

### 4.1. Cell Culture and Chemical Treatment

Mouse embryonic fibroblast (MEF) cells were kindly provided by Dr. Randal J. Kaufman. Cells were maintained in DMEM (Corning, Corning city, NY, USA) supplemented with 10% (*v*/*v*) heat-inactivated fetal bovine serum (FBS, Corning) and 1% (*v*/*v*) penicillin and streptomycin (P/S, Corning) at 37 °C in an incubator with 5% CO_2_. Cells were treated with thapsigargin (TG; Sigma Aldrich, St. Louis, MO, USA), 4μ8c (Cayman, Ann Arbor, MI, USA, 22110), or ceapinA7 (Sigma Aldrich) at the indicated concentrations and time points. MEFs expressing Fv2E-PERK were treated with AP20187 (B/B homodimerizer; Takara, Kusatsu, Shiga, Japan) at the indicated concentrations and time points.

### 4.2. Western Blotting

Protein lysates were prepared in RIPA lysis buffer (Pierce, Waltham, MA, USA) supplemented with Halt protease and phosphatase inhibitor cocktail (Thermo Scientific, Waltham, MA, USA). Protein concentrations were measured using a BCA assay kit (Pierce). Ten to 20 μg of protein was mixed with Laemmli sample buffer (Bio-Rad, Hercules, CA, USA) containing 10% (*v*/*v*) 2-mercaptoethanol (Sigma Aldrich) and incubated at 100 °C for 5 min before loading onto an SDS-PAGE gel. The blots were probed with primary antibodies as follows: GADD45α (Santa Cruz Biotechnology, Dallas, TX, USA, sc-797), cyclin A (Santa Cruz Biotechnology, sc-751), cyclin B1 (Santa Cruz Biotechnology, sc-245), ATF4 (Cell Signaling Technology, Danvers, MA, USA, #11815), CHOP (GADD153; Santa Cruz Biotechnology, sc-7351), phospho-eIF2α (Abcam, Cambridge, England, ab32157), eIF2α (Cell Signaling Technology, #9722), Caspase-3 (Cell Signaling Technology, #9662), cleaved Caspase-3 (Cell Signaling Technology, #9661), PARP (Cell Signaling Technology, #9542), and α-Tubulin (Sigma, St. Louis, MO, USA, T9026). All the antibodies used in the experiments were diluted according to the manufacturer’s recommendations. Antibody signals were detected on the imaging film using an automatic film processor (Agfa, Mortsel, Belgium). The band intensity was quantified by ImageJ software (National Institutes of Health, NIH, Bethesda, MD, USA, version 1.52a).

### 4.3. Cell Cycle Analysis

Determination of the cell cycle population was quantified using propidium iodide (PI) staining. Cells were fixed and permeabilized with 70% ethanol at −20 °C for overnight. Fixed cells were washed with PBS twice, resuspended with 20 μg/mL of RNase A in PBS, and stained with a final concentration of 10 μg/mL PI in the dark. Cells were subjected to analysis using a BD FACS canto (Becton Dickinson, Franklin Lakes, NJ, USA) and Flowjo software (Becton Dickinson).

### 4.4. siRNATransfection

For silencing the expression of GADD45α, scrambled siRNAs for non-target control and pre-designed GADD45α siRNAs (Bioneer, Daejeon, Korea) were used. Transfection in wildtype MEF cells was performed using MATra-si reagent (IBA Lifesciences, Göttingen, Germany) according to the manufacturer’s instructions, with some modifications. Briefly, 6 μL of MATra-si reagent and a final concentration of 100 nM of siRNA were incubated with 200 μL of Opti-MEM medium (Thermo Scientific) at room temperature as around 23 to 27 °C for 20 min. The siRNA mixture was then added to 30% of the cells and placed on a magnetic plate to incubate for 15 min. The knockdown efficiency was verified by western blotting performed 48 h after transfection, and then 8 h after TG treatment with the negative control. GADD45α siRNA sequences used in this study were as follows: Sense; GCAAUAUGACUUUGGAGGA(dTdT), Antisense; UCCUCCAAAGUCAUAUUGC(dTdT).

### 4.5. RNA Extraction and Quantitative Real Time PCR (qRT-PCR)

Total RNA was extracted from cells using Ribo-EX (GeneAll, Seoul, Korea), and cDNA was synthesized using CellScript cDNA master mix (Cellsafe, Yongin, Korea). The relative amounts of mRNA were calculated from the comparative threshold cycle (Ct) values relative to *Gapdh* using CFX96 Real-Time PCR detection system (Bio-Rad, 184-5384) with SYBR green reagent (Enzynomics, Daejeon, Korea) according to the manufacturer’s instructions. Real-time primer sequences used in this study were as follows: *Gadd45α* (5’-TGC TAC TGG AGA ACG ACG C-3’; 5’-GGA TCC TTC CAT TGT GAT GAA-3’), *Chop* (5′-CTG CCT TTC ACC TTG GAG AC-3′; 5′-CGT TTC CTG GGG ATG AGA TA-3′), spliced *Xbp1* (5′-GAG TCC GCA GCA GGT G-3′; 5′-GTG TCA GAG TCC ATG GGA-3′), total *Xbp1* (5′-AAG AAC ACG CTT GGG AAT GG-3′; 5′-ACT CCC CTT GGC CTC CAC-3′), *BiP* (5′-GGT GCA GCA GGA CAT CAA GTT-3′; 5′-CCC ACC TCC AAT ATC AAC TTG A-3′), *p58* (5’-TCC TGG TGG ACC TGC AGT ACG-3’; 5’-CTG CGA GTA ATT TCT TCC CC-3’), *Ero1l* (5’-CAC AGG TAC AGT CGT CCA GGT-3’; 5’-CTT GCT CGT TGG ACT CCT G-3’), and *Herpud1* (5’-CAA CAG CAG CTT CCC AGA AT-3’; 5’-CCG CAG TTG GAG TGT GAG T-3’).

### 4.6. Bioinformatic Analysis

Gene expression data for MEFs with ER stress were obtained from the Gene Expression Omnibus (GSE35681). Differentially expressed genes (DEGs) were defined as genes with a statistically significant expression change (FDR < 0.05 with a fold-change over 1) in wild-type MEFs exposed to ER stress compared to control MEFs, and were clustered by Gene ontology (GO) analysis using bioinformatics resources from the Protein Analysis Through Evolutionary Relationships (PANTHER) Classification System 14.1 (http://www.pantherdb.org/about.jsp). Enriched values were presented as their biological process (BP) categories. Upregulated *Gadd45α* was visualized by a heatmap using Prism software (Version 7.05).

### 4.7. Statistical Analysis

Data are presented as means ±SEM. Unpaired Student’s t test for single comparison and two-way ANOVA for multiple comparison of FACS analysis were used to assess the statistical significance of the differences between groups. P-values less than 0.05 were considered statistically significant compared to the negative control. Experiments were repeated at least three times.

## Figures and Tables

**Figure 1 ijms-20-06309-f001:**
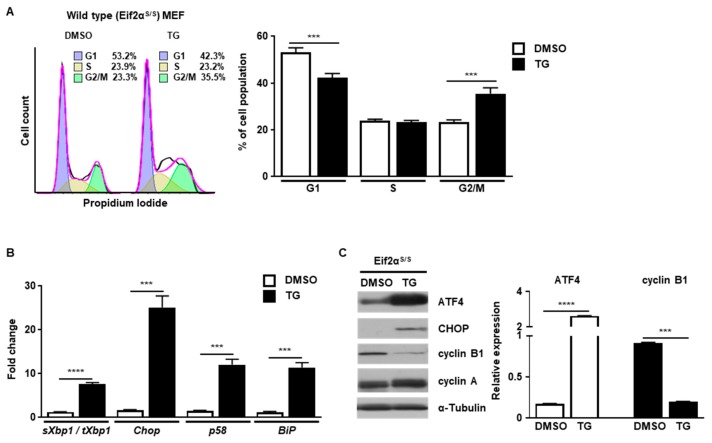
ER stress induces cell cycle arrest at the G2/M phase. Wild type (Eif2α^S/S^) MEFs were collected at 8 h following treatment with DMSO or thapsigargin (TG; 300 nM) for FACS analysis (**A**), quantitative RT-PCR (**B**) or western blotting (**C**). Percentages of cell populations are presented as means (*n* = 3). *Gapdh* primers were used as an endogenous control for quantitative RT-PCR. α-Tubulin was used as an endogenous control for western blotting. Normalized band densities were quantified using ImageJ software. *** *p* < 0.005; **** *p* < 0.001.

**Figure 2 ijms-20-06309-f002:**
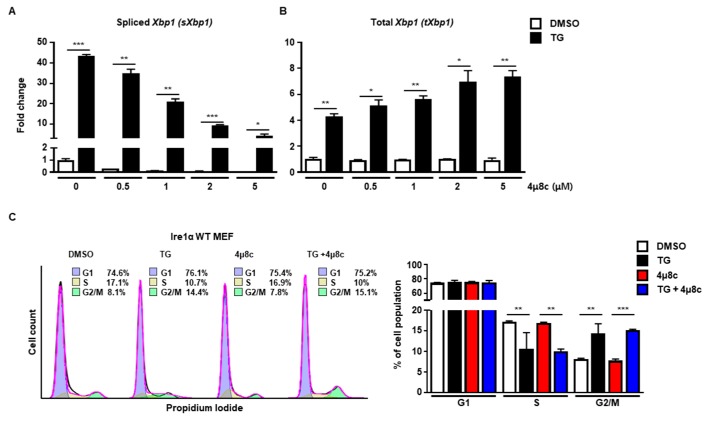
The IRE1α pathway is not involved in G2/M cell cycle arrest induced by ER stress. (**A**,**B**) Wild type (Ire1α WT) MEFs were collected at 8 h following treatment with DMSO or thapsigargin (TG; 300 nM) in the presence or absence of 4μ8c at indicated doses for quantitative RT-PCR. *Gapdh* primers were used as an endogenous control. (**C**) Ire1α WT MEFs were collected at 8 h following treatment with DMSO or TG (300 nM) in the presence or absence of 4μ8c (1 μM) for FACS analysis. Percentages of cell populations are presented as means (*n* = 3). * *p* < 0.05, ** *p* < 0.01, *** *p* < 0.005.

**Figure 3 ijms-20-06309-f003:**
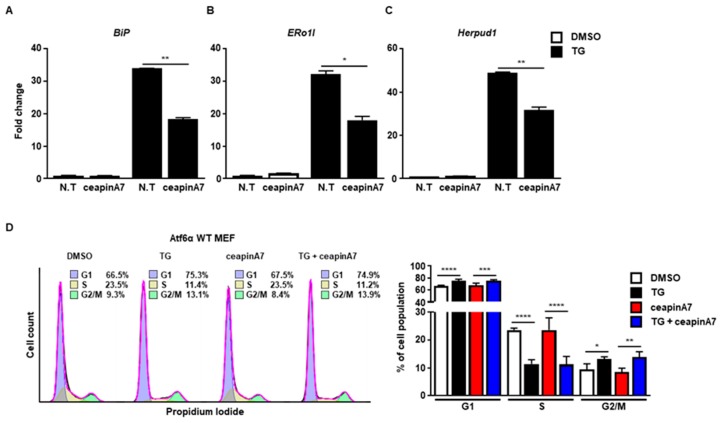
The ATF6α pathway is not involved in G2/M cell cycle arrest caused by ER stress. Wild type (Atf6α WT) MEFs were collected at 8 h following treatment with DMSO or thapsigargin (TG; 300 nM) in the presence or absence of ceapinA7 (0.5 μM) for quantitative RT-PCR (**A**–**C**), or FACS analysis (**D**). *Gapdh* primers were used as an endogenous control for quantitative RT-PCR. Percentages of cell populations are presented as means (*n* = 4). * *p* < 0.05, ** *p* < 0.01, *** *p* < 0.005, **** *p* < 0.001

**Figure 4 ijms-20-06309-f004:**
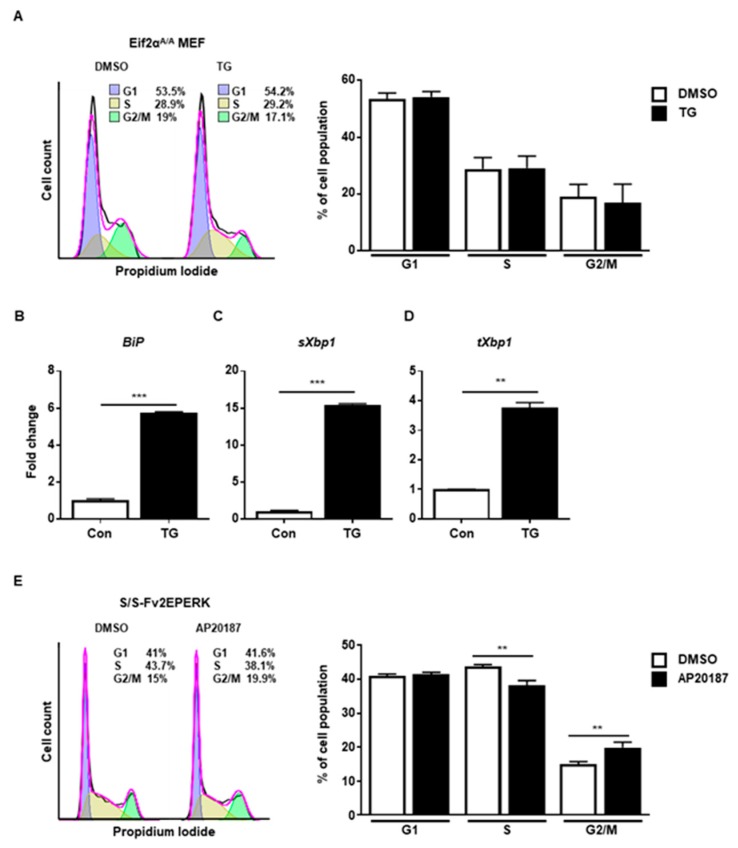
The PERK-eIF2α signaling pathway is involved in G2/M cell cycle arrest. (**A**–**D**) eIF2αSer51Ala mutant (Eif2α^A/A^) MEFs were collected at 8 h following treatment with DMSO or thapsigargin (TG; 300 nM) for FACS analysis (**A**) or quantitative RT-PCR (**B**–**D**). Percentages of cell populations are presented as means (*n* = 3). *Gapdh* primers were used as an endogenous control for quantitative RT-PCR. (**E**) Wild-type MEFs expressing Fv2E-PERK (S/S-Fv2EPERK) were collected at 8 h following treatment with DMSO or AP20187 (1 nM) for FACS analysis. Percentages of cell populations are presented as means (*n* = 4). ** *p* < 0.01, *** *p* < 0.005.

**Figure 5 ijms-20-06309-f005:**
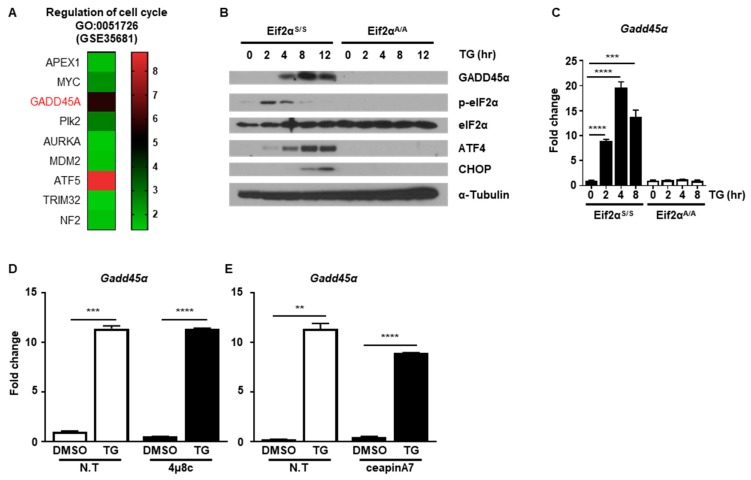
The PERK-eIF2α signaling pathway induces GADD45α expression. (**A**) Heatmap of the gene ontology (GO) enriched genes belong to the term “regulation of cell cycle” (GO:0051726). (**B**) Cell lysates were obtained from wild type (Eif2α^S/S^) or eIF2αSer51Ala mutant (Eif2α^A/A^) MEFs at the indicated time points following treatment with DMSO or thapsigargin (TG; 300 nM) (n = 3). α-Tubulin was used as an endogenous control. (**C**) Total RNAs were obtained from Eif2α^S/S^ or Eif2α^A/A^ MEFs at the indicated time points following treatment with DMSO or TG (300 nM) (*n* = 3). *Gapdh* primers were used as an endogenous control. (**D**,**E**) Total RNAs were obtained from wild type MEFs at 8 h following treatment with DMSO or TG (300 nM) in the presence or absence of 4μ8C (1 μM) (**D**) or ceapinA7 (0.5 μM) (**E**) (*n* = 3). *Gapdh* primers were used as an endogenous control. ** *p* < 0.01, *** *p* < 0.005.

**Figure 6 ijms-20-06309-f006:**
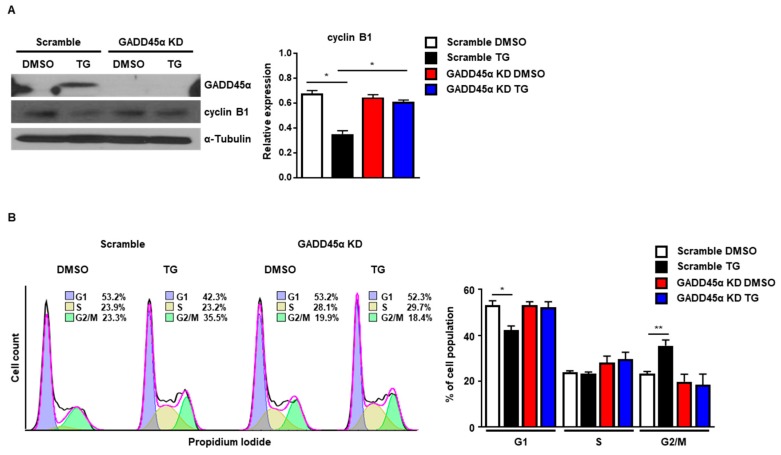
Knockdown of GADD45α-impaired G2/M cell cycle arrest caused by ER stress. Wild-type MEFs were transduced with either scramble control siRNA or GADD45α siRNA (100 nM) for 48 h. After 8 h with DMSO or thapsigargin (TG; 300 nM) treatment, cells were collected for western blotting (**A**) or FACS analysis (**B**) (*n* = 3). α-Tubulin was used as an endogenous control for western blotting. * *p* < 0.05, ** *p* < 0.01.

**Figure 7 ijms-20-06309-f007:**
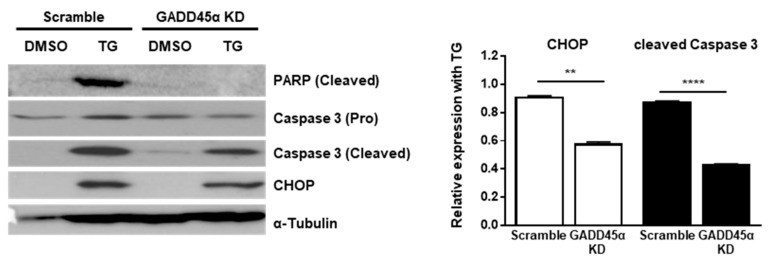
Knockdown of GADD45α-impaired cell death caused by ER stress. Wild type MEFs were transduced with either scramble control siRNA or GADD45α siRNA (100 nM) for 48 h. After 12 h with DMSO or TG (300 nM) treatment, cells were collected for western blotting (*n* = 3). α-Tubulin was used as an endogenous control. Normalized band densities were quantified using ImageJ software. ** *p* < 0.01; **** *p* < 0.001.

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
