# Peer review of "ER Stress Induces Cell Cycle Arrest at the G2/M Phase Through eIF2α Phosphorylation and GADD45α"

_ijms, 2019, doi:10.3390/ijms20246309_

Round 1

Reviewer 1 Report

Overall it is a well organized work but some minor comments to improve the manuscript:

Figure 1A and 2C (and other figures too): for consistency, write ‘DMSO’ instead of ‘Con’ in the flow cytometry (FACS) figure.

Also state what the color codes refer to in FACS data.

Mention which statistics test was performed to indicate the significance (p) values.

Lines 103-104: mention that spliced Xbp1 is sXbp1 and total Xbp1 is tXbp1 in figure 2A.

Reviewer 2 Report

Paper presents results of experiments analyzing ER stress as an inductor of cell cycle arrest at G2_M phase through eIF 2alpha phosphorylation and GADD45alpha. Authors on the basis of their experiments concluded that GADD alpha plays a key role in the ER stress induced cell cycle arrest. Authors also noted that this phenomenon is linked with the eIF 2 signalling pathway. The paper is  very interesting and it addresses the problem that has not been fully solved yet and is related to the lack of ethiology of cell death induction. Design of the study is well prepared. Although the novelty of this study is not topical, reflected the large number of reports published in this topic, but the methodological approach and carefull analysis of data makes it very valueable.

There are some points, however to address:

the length of the manuscript seems to be shorten, especially the Discussion References can be concentrated to the most relevant ones Figure legends seems to be quite wordy and some sentenses already written in the former captures there are some typing errors to be corrected  and grammar and style needs minor revision

Reviewer 3 Report

In the manuscript entitled "ER stress induces cell cycle arrest at the G2/M phase  through eIF2α phosphorylation and GADD45α" Lee et al. investigate the role of GADD45α in ER stress-induced G2/M cell cycle arrest in mouse embryonic fibroblasts (MEFs). Overall, I think that Authors made an effort to conduct this research, but I also have some concerns regarding this manuscript. My detailed comments are given below:

In the Introduction/Discussion, some emphasis should be put into the possible practical use of the investigated phenomenon e.g. in context of cell type used in this study. Why MEFs were chosen for the examination? I understand that Authors focused on investigating cell cycle arrest in this study, however, I am wondering why apoptosis was not demonstrated here. It would be interesting to see what is the relationship between apoptotic/cell cycle arrested cells, especially knowing that TG is a strong inducer of ER stress and apoptosis seems to be its primary mode of action. Why was 8 hours chosen as a final time of incubation for almost all investigations? Longer time point would result in apoptosis? How would Authors know to choose the right time without first checking at least several time points? Figure 1C, why was only cyclin B1 subjected to densitometry analysis, while (according to the corresponding WB analysis) there are obvious differences in the expression of other proteins too? Figure 2A – a graph without ,,DMSO columns’’ looks weird. Maybe some readjustments in the scale? Line 142-144 it is stated: ,,When Fv2E-PERK was dimerized and activated by AP20187, cell cycle arrest at the G2/M phase was clearly observed compared to the  control at 8 hours (Figure 4E)’’. I don’t know if it can be stated this way, since this ,,clear observation’’ is in fact only 4,9 %  difference? Please explain how was data concerning gene expression and GO ontology extracted? Was it based on the microarray analysis performed by Authors themselves? Why is TM and not TG used as ER stressor? WB image of GADD45α in Fig.5 looks like full of artefacts. Please replace with different one. The layout of paragraphs 2.4 and 2.5 looks weird since they both refer to the same Figure 6, just different panels of it. Please rearrange in a way be more clear. Also, in Fig.6, why 8 h treatment with TG (Panels A, B) and 12 h treatment with TG (Panel C) was applied? What is the rationale behind using different time points here? Last phrase of the Discussion (lines 281-283) Therefore, pharmacological agents that target Gadd45α may be  helpful for the treatment of ER stress-associated diseases,  including type Ⅱ diabetes [59], neurodegenerative diseases [60], and cancer [61].’’ I think that this sentence is too superficial and poorly thought through. How can ,,targeting’’ Gadd45α benefit e.g. diabetes and cancer? In case of both disorders the focus of such targeting would be completely different (in first to alleviate ER stress and in second to augment ER stress to the point of apoptotic cell death). This should be teased more carefully. Please make some deeper analysis. What is the key of using different notation of GADD45α vs. Gadd45α?

Round 2

Reviewer 3 Report

Manuscrtipt has been significantly improved.